# Assisted cloning of an unknown shared quantum state

**Dengxin Zhai**[ID][‡], **Jiayin Peng**[ID][‡]*, **Nueraminaimu Maihemuti**[‡], **Jiangang Tang**[‡]

School of Mathematics and Statistics, Kashi University, Kashi, Xinjiang, China

‡ These authors contributed equally to this work.
‡ NM and JT also contributed equally to this work.
* pengjiayin62226@163.com

**Data Availability Statement:** All relevant data are within the article.

**Funding:** This work is supported by the Kashi University Flexible Introduction Research Initiation Fund (No. 022024077). The funder was involved in

## Abstract

We first propose a novel protocol to realize quantum cloning of an arbitrary unknown shared state with assistance offered by a state preparer. The initial phase of this protocol involves the utilization of quantum teleportation (QT), enabling the transfer of quantum information from an arbitrary number of senders to another arbitrary number of receivers through a maximally entangled GHZ-type state serving as a network channel, without centralizing the information at any specific location. In the second stage of this protocol, the state preparer performs a special single-qubit projective measurement and multiple Z-basis measurements and then communicates a number of classical bits corresponding to measurement results, the perfect copy or orthogonal-complementing copy of an unknown shared state can be produced at senders hands. Then, using a non-maximally entangled GHZ-type state instead of the aforementioned quantum channel, we extend the proposed protocol from three perspectives: projective measurement, positive operator-value measurement (POVM), and a single generalized Bell-state measurement. Our schemes can relay quantum information over a network without requiring fully trusted central or intermediate nodes, and none of participants can fully access the information.

## 1 Introduction

The quantum teleportation (QT) scheme proposed by Bettnett et al. [1] in 1993 pioneered quantum information science which today is a vast research field. Its superior potential for application is undisputed. Especially, it is a crucial task in the implementation of quantum networks for promising applications such as quantum cryptography [2] and distributed quantum computation [3, 4]. Although the original QT scheme teleports quantum information from one place to another [1], incorporation of multiple participants is further worth considering to implement versatile quantum networks. Schemes to share quantum information from one sender to multiple receivers have been presented [5–8] and experimentally demonstrated [9, 10] via the multipartite entanglement state serve as quantum channel. In these schemes, no single receiver or any subparties of receivers can fully access information unless all other receivers cooperate, which forms the basis for further expanding quantum secret sharings [11–14] or controlled teleportations [15, 16]. Besides the aforementioned unidirectional QT,

the research design, data collection and analysis, publication decision, or manuscript preparation.

**Competing interests:** The authors have declared that no competing interests exist.

bidirectional QT [17, 18] and cyclic QT [19–21] have been studied. Furthermore, there exist other typical quantum communication protocols that can facilitate the establishment of versatile quantum networks [22–28].Various quantum cryptography schemes, such as quantum key distribution [22, 23], quantum secure direct communication [24, 25] and remote state preparation (RSP) [26, 27], are capable of establishing a secure communication channel. In these schemes, eavesdropping is impossible without a high probability of interfering with transmission, as eavesdropping will be detected.

Unlike QT which uses pre-shared quantum entangled channel and classical communication to teleport unknown quantum state, RSP is utilized to teleport a known quantum state, which can save communication resources compared to QT [29, 30]. Various RSP schemes have emerged, such as multicast-based multiparty RSP [31], controlled RSP [32, 33], joint RSP [34, 35], controlled joint RSP [36], bidirectional controlled RSP [37, 38], cyclic RSP [39], etc. In 2020, Lee et al. [40] introduced a novel QT scheme enabling the transfer of quantum information from an arbitrary number of senders to another arbitrary number of receivers in an efficient and distributed manner over a network, without the need for fully trusted central or intermediate nodes. Furthermore, this scheme can be extended to include error corrections for photon losses, bit or phase-flip errors, and dishonest parties. This work paves the way for secure distributed quantum communications and computations in quantum networks. In 2022, Li et al. [41] extended Lee's scheme [40] to the case of non-maximally entangled channel.

To manipulate and extract quantum information, Pati [42] proposed a scheme in 2000 using QT and RSP techniques to generate perfect copies and orthogonal-complement copies of an arbitrary unknown state with minimal assistance from the state preparer. The first stage of this scheme requires the usual teleportation, while in the second stage the preparer carries out a single-qubit measurement and conveys some classical information to different parties so that perfect copies and orthogonal-complement copies are produced in a probabilistic manner. Zhan [43] proposed a scheme of realizing quantum cloning of an unknown two-particle entangled state and its orthogonal complement state with assistance from a state preparer. Han et al. [44] presented a scheme that can clone an arbitrary unknown two-particle state and its orthogonal complement state with the assistance of a state preparer, where a genuine four-particle entangled state is used as the quantum channel and positive operator-valued measurement (POVM) instead of usual projective measurement is employed. In Ref. [45], by using a non-maximally four-particle cluster state as quantum channel, a scheme for cloning unknown two-particle entangled state and its orthogonal complement state with assistance from a state preparer was proposed. Zhan et al. [46] proposed a protocol where one can realize quantum cloning of an unknown two-particle entangled state and its orthogonal complement state with assistance offered by a state preparer. The following year, Fang et al. [47] generalize Zhan's protocol [46] such that an arbitrary unknown two-qubit entangled state can be treated. Zhan [48] presented a scheme for realizing assisted cloning of an unknown two-atom entangled state via cavity QED. Hou and Shi suggested a protocol for cloning an unknown EPR-type state with assistance by using a one-dimensional non-maximally four-particle cluster state as quantum channel, and then extend it to the case of cloning an arbitrary unknown two-particle entangled state. Hou and Shi [45] suggested a protocol for cloning an unknown EPR-type state with assistance by using a one-dimensional non-maximally four-particle cluster state as quantum channel, and then extend it to the case of cloning an arbitrary unknown two-particle entangled state. Xiao et al. [49] put forward a protocol of assisted cloning and orthogonal complementing of an arbitrary two-qubit state via two partially entangled pairs as quantum channel. Shi et al. [50] proposed a protocol which can realize quantum cloning of an unknown tripartite entangled state and its orthogonal complement state with assistance from a state preparer. Ma et al. [51] present a scheme which can realize quantum cloning of an unknown *N*-

particle entangled state and its orthogonal complement state with assistance offered by a state via $N$ non-maximally entangled particle pairs as quantum channel. Subsequently, they proposed a scheme to produce perfect copy of an unknown $d$-dimensional equatorial quantum state with assistance offered by a state preparer [52]. Chen et al. [53] and Xue et al. [54] extended the scheme [52] to the cases for auxiliary cloning of an unknown multi-qudit equatorial-like state and an arbitrary unknown multi-qudit state, respectively.

Stimulated by Refs. [40–42], in this manuscript, we explore the assisted cloning of shared quantum secret state. We first propose a new scheme for cloning an unknown shared quantum state or its orthogonal complement state with help of a state preparer. This scheme includes two stages: teleportation and assisted cloning. The first stage of the scheme requires quantum teleportation, which uses a maximally entangled GHZ-type state as the network channel to teleport an arbitrary unknown shared quantum state between multiple senders and receivers. In the second state of the scheme, the preparer disentangles the left over entangled states by a single-qubit projective measurement and some $Z$-basis measurements and informs senders of his or her measurement outcomes so that perfect copies and complement copies of the unknown shared state are producd. In addition, we discuss the assisted cloning problem of shared quantum secret from three perspectives: projective measurement, POVM, and a single generalized Bell-state measurement, by replacing the aforementioned network channel with a non-maximally entangled GHZ-type state. The results show that the obtained cloning schemes are all extensions of the above scheme, and they can achieve unit fidelity, but the cost is that the success probability is less than 1.

Below is the arrangement of this article. In Section 2, by using a maximally entangled GHZ-type state as quantum channel, a new scheme for cloning an arbitrary unknown shared state and its orthogonal complement state with the assistance from a state preparer is presented. In Section 3, by making use of a non-maximally entangled state as network channel, we provide three assisted cloning schemes of shared quantum secret via projective measurement, POVM, and a single generalized Bell-state measurement, respectively, to meet the needs of real environments and the purpose of expanding the scheme in Section 2. Discussion and conclusion are drawn in Section 4.

## 2 Assisted cloning of shared quantum secret via a maximally entangled GHZ-type state

Suppose that Victor is the preparer of the quantum state $|\chi\rangle = \alpha|0\rangle + \beta|1\rangle$, where $\alpha, \beta$ are complex numbers with $|\alpha|^2 + |\beta|^2 = 1$. A quantum secret in $|S\rangle = \alpha|0_L\rangle + \beta|1_L\rangle$ with logical basis, $|0_L\rangle$ and $|1_L\rangle$, is shared by separated $n$ parties $\{A_1, A_2, \cdots, A_n\}$ in quantum network, through a splitting protocol [8–11], where the the state $|S\rangle$ can be rewritten as

$$|S\rangle_s^n = \alpha \otimes_{l=1}^n |0\rangle_{s_l} + \beta \otimes_{l=1}^n |1\rangle_{s_l}, \tag{1}$$

where qubit $s_j$ belongs to sharer $A_j$ ($j = 1, 2, \cdots, n$). That is to say, state $|S\rangle_s^n$ is the result of state $|\chi\rangle$ being shared by $n$ individuals $\{A_1, A_2, \cdots, A_n\}$ in the network. Our utilization of GHZ-entanglement of photons enables the encoding of network and logical qubits. The senders $\{A_1, A_2, \cdots, A_n\}$, i.e., a group of n parties, endeavor to transmit the shared secret to the receivers, i.e., another group $\{B_1, B_2, \cdots, B_m\}$ of m parties interconnected in the network, and the shared secret state is obtained at the receivers' hands could be reconstructed as $|S\rangle_r^m = \alpha \otimes_{j=1}^m |0\rangle_{r_j} + \beta \otimes_{j=1}^m |1\rangle_{r_j}$, here qubit $r_j$ belongs to receiver $B_j$ ($j = 1, 2, \cdots, m$), and then wish to clone this shared secret state at the senders' hands with the assistance from the state preparer Victor. None of the participants are fully trusted, therefore no single sender or receiver, or any subparties, is permitted to access the secret during the entire process.

In order to accomplish this objective, the network channel utilizes an $(n + m)$-particle GHZ-type state

$$|\mathcal{G}\rangle_{n+m} = \frac{1}{\sqrt{2}}(\otimes_{l=1}^{n}|0\rangle_{s_l'}\otimes_{j=1}^{m}|0\rangle_{r_j} + \otimes_{l=1}^{n}|,1\rangle_{s_l'}\otimes_{j=1}^{m}|1\rangle_{r_j}) \tag{2}$$

where qubit $s_l'$ belongs to sender $A_l$ ($l = 1, 2, \cdots, n$), while the channel particle $r_j$ is also owned by receiver $B_j$ ($j = 1, 2, \cdots m$). The assisted cloning scheme between multiple parties in a quantum network includes two stages: quantum teleportation and copying of unknown state, and the specific process is presented below.

In the first stage of the scheme, each sender $A_j$ executes the standard Bell-state measurement on her or his two qubits $s_j$ and $s_j'$, one from $|S\rangle_s^n$ and the other from the network channel $|\mathcal{G}\rangle_{n+m}$. The Bell-states can be represented as

$$|\varphi^{\pm}\rangle \frac{1}{\sqrt{2}}(|00\rangle \pm |11\rangle), \quad |\psi^{\pm}\rangle \frac{1}{\sqrt{2}}(|01\rangle \pm |10\rangle). \tag{3}$$

Utilizing the aforementioned Bell-states, the initial composite system can be expressed as follows

$$\begin{aligned}
|S\rangle_s^n|\mathcal{G}\rangle_{n+m} &= (\alpha\otimes_{l=1}^{n}|0\rangle_{s_l} + \beta\otimes_{l=1}^{n}|1\rangle_{s_l}) \\
&\otimes\frac{1}{\sqrt{2}}(\otimes_{l=1}^{n}|0\rangle_{s_l'}\otimes_{j=1}^{m}|0\rangle_{r_j} + \otimes_{l=1}^{n}|1\rangle_{s_l'}\otimes_{j=1}^{m}|1\rangle_{r_j}) \\
&= \frac{1}{2}|\Phi_n^{+}\rangle_{s_1\cdots s_n s_1'\cdots s_n'}(\alpha\otimes_{j=1}^{m}|0\rangle_{r_j} + \beta\otimes_{j=1}^{m}|1\rangle_{r_j}) \\
&+ \frac{1}{2}|\Phi_n^{-}\rangle_{s_1\cdots s_n s_1'\cdots s_n'}(\alpha\otimes_{j=1}^{m}|0\rangle_{r_j} - \beta\otimes_{j=1}^{m}|1\rangle_{r_j}) \\
&+ \frac{1}{2}|\Psi_n^{+}\rangle_{s_1\cdots s_n s_1'\cdots s_n'}(\alpha\otimes_{j=1}^{m}|1\rangle_{r_j} + \beta\otimes_{j=1}^{m}|0\rangle_{r_j}) \\
&+ \frac{1}{2}|\Psi_n^{-}\rangle_{s_1\cdots s_n s_1'\cdots s_n'}(\alpha\otimes_{j=1}^{m}|1\rangle_{r_j} - \beta\otimes_{j=1}^{m}|0\rangle_{r_j}),
\end{aligned} \tag{4}$$

where

$$\begin{aligned}
|\Phi_n^{\pm}\rangle_{s_1\cdots s_n s_1'\cdots s_n'} &= \frac{1}{\sqrt{2}}(\otimes_{l=1}^{n}|0\rangle_{s_l}\otimes_{l=1}^{n}|0\rangle_{s_l'} \pm \otimes_{l=1}^{n}|1\rangle_{s_l}\otimes_{l=1}^{n}|1\rangle_{s_l'}), \\
|\Psi_n^{\pm}\rangle_{s_1\cdots s_n s_1'\cdots s_n'} &= \frac{1}{\sqrt{2}}(\otimes_{l=1}^{n}|0\rangle_{s_l}\otimes_{l=1}^{n}|1\rangle_{s_l'} \pm \otimes_{l=1}^{n}|1\rangle_{s_l}\otimes_{l=1}^{n}|0\rangle_{s_l'}).
\end{aligned} \tag{5}$$

The arrangement of $2n$ particles is modified from $(s_1, s_2, \cdots, s_n, s_1', s_2', \cdots, s_n')$ to $(s_1, s_1', s_2, s_2', \cdots, s_n, s_n')$ and we have (for brevity, the subscripts are omitted)

$$\begin{aligned}
|\Phi_n^{+}\rangle &= \frac{1}{\sqrt{2^{n-1}}}\sum_{0\leq k=0\bmod 2\leq n}N[|\phi^{-}\rangle^{\otimes k}|\phi^{+}\rangle^{\otimes(n-k)}], \\
|\Phi_n^{-}\rangle &= \frac{1}{\sqrt{2^{n-1}}}\sum_{0\leq k=1\bmod 2\leq n}N[|\phi^{-}\rangle^{\otimes k}|\phi^{+}\rangle^{\otimes(n-k)}], \\
|\Psi_n^{+}\rangle &= \frac{1}{\sqrt{2^{n-1}}}\sum_{0\leq k=0\bmod 2\leq n}N[|\psi^{-}\rangle^{\otimes k}|\phi^{+}\rangle^{\otimes(n-k)}], \\
|\Psi_n^{+}\rangle &= \frac{1}{\sqrt{2^{n-1}}}\sum_{0\leq k=1\bmod 2\leq n}N[|\psi^{-}\rangle^{\otimes k}|\phi^{+}\rangle^{\otimes(n-k)}],
\end{aligned} \tag{6}$$

in which $N[\cdot]$ represents the sum of all possible arrangements, for example,

$$
\begin{aligned}
N[|\phi^-\rangle|\phi^+\rangle^{\otimes 3}] \quad &= |\phi^-\rangle|\phi^+\rangle|\phi^+\rangle|\phi^+\rangle + |\phi^+\rangle|\phi^-\rangle|\phi^+\rangle|\phi^+\rangle \\
&+ |\phi^+\rangle|\phi^+\rangle|\phi^-\rangle|\phi^+\rangle + |\phi^+\rangle|\phi^+\rangle|\phi^+\rangle|\phi^-\rangle.
\end{aligned}
\tag{7}
$$

Upon conducting $n$ Bell-state measurements, they communicate the outcomes to the recipients through classical channels. As a priori agreement, define the measurement result of $A_l$ as $x_l y_l$ and let the classical bits 00, 01, 10 and 11 correspond to the Bell-states $|\phi^+\rangle$, $|\phi^-\rangle$, $|\psi^+\rangle$ and $|\psi^-\rangle$, respectively, and vice vera.

If the measurement outcome is $|\Phi_n^+\rangle$ or $|\Phi_n^-\rangle$, at any receiver's location, he performs the local Pauli operator $\sigma_z^X$, where $\sigma_z = |0\rangle\langle 0| - |1\rangle\langle 1|$ and $X = (\sum_{l=1}^n x_l) \bmod 2$. After the above operations, the state of qubits $r_1, r_2, \cdots, r_{m-1}$ and $r_m$ becomes

$$
|S\rangle_{r_1 r_2 \cdots r_m} = \alpha \otimes_{j=1}^m |0\rangle_{r_j} + \beta \otimes_{j=1}^m |1\rangle_{r_j},
\tag{8}
$$

which means that the receivers $B_1, B_2, \cdots, B_m$ successfully reconstruct the shared initial state $|S\rangle_s^n$ with unit fidelity. If the measurement result from the senders is either $|\Psi^+\rangle$ or $|\Psi^-\rangle$, one of the receivers will apply the local Pauli operator $\sigma_x^Y \sigma_z^X$, where $\sigma_x = |0\rangle\langle 1| + |1\rangle\langle 0|$ and $Y = \sum_{l=1}^n y_l$, while the other receivers respectively carry out the operator $\sigma_x^Y$. This will result in the state owned by the receivers becoming $|S\rangle_{r_1 r_2 \cdots r_m}$ as shown in Eq (8). That is, the receivers can successfully restore the shared initial state $|S\rangle_s^n$ with unit fidelity, completing teleportation.

Now we move on to the second stage of the scheme: creating a copy or an orthogonal-complementing copy of the unknown state $|S\rangle_s^n$ with assistance from the state preparer. According to the projection postulate of quantum mechanics, without loss of generality, If the senders' Bell measurement result is $(|\phi^+\rangle\langle\phi^+|)^{\otimes n}$, the state of qubits $s_1, s_1', s_2, s_2', \cdots, s_n$ and $s_n'$ will collapse into the state $|\phi^+\rangle^{\otimes n}$, (see Eqs (4), (5) and (6)). Each sender $A_i$ ($i = 1, 2, \cdots, n$) sends qubit $s_i$ to the state preparer Victor and keeps qubit $s_i'$ in his or her possession. Since Victor knows the state $|\chi\rangle$ completely, he performs a single-qubit projective measurement on the qubit $s_1$ in a set of mutually orthogonal basis vectors $\{|\xi_0\rangle_{s_1}, |\xi_1\rangle_{s_1}\}$, which is given by

$$
|\xi_0\rangle_{s_1} = \alpha|0\rangle_{s_1} + \beta|1\rangle_{s_1}, \quad |\xi_1\rangle_{s_1} = \alpha^*|1\rangle_{s_1} - \beta^*|0\rangle_{s_1}.
\tag{9}
$$

Subsequently, he measures each other qubit using the $Z$-basis $\{|0\rangle, |1\rangle\}$, and publishes the measurement results to the senders through classical communication.

Using Victor's measurement bases $\{|\xi_0\rangle_{s_1}, |\xi_1\rangle_{s_1}\}$ and $\{|0\rangle, |1\rangle\}$, $|\phi^+\rangle^{\otimes n}$ can be written as

$$
|\phi^+\rangle^{\otimes n} = \frac{1}{\sqrt{2^n}} \left[ |\xi_0\rangle_{s_1} (\alpha^*|0\rangle + \beta^*|1\rangle)_{s_1'} + |\xi_1\rangle_{s_1} (\alpha|1\rangle - \beta|0\rangle)_{s_1'} \right] \otimes_{i=2}^n \sum_{k_i=0}^1 |k_i\rangle_{s_i} |k_i\rangle_{s_i'}.
\tag{10}
$$

Obviously, Eq (9) is a transitional transformation from the old basis $\{|0\rangle, |1\rangle\}$ to the new basis $\{|\xi_0\rangle, |\xi_1\rangle\}$. It is worth noting that under this transformation, the normalization and orthogonality relationships between the basis vectors are preserved. Interestingly, we find the basis vector $|\xi_0\rangle = |\chi\rangle$ and the basis vector $|\xi_1\rangle = |\chi_\perp\rangle$, where $|\chi_\perp\rangle = \alpha^*|1\rangle - \beta^*|0\rangle$ is the orthogonal-complement state to $|\chi\rangle$. However, we keep $|\xi_0\rangle_{s_1}, |\xi_1\rangle_{s_1}$ for Victor just to distinguish the fact that he knows the state.

Generally speaking, if Victor's measurement results for particle $s_1$ and particle $s_j$ ($j = 2, 3, \cdots, n$) are $|\xi_t\rangle_{s_1}$ ($t = 0, 1$) and $|k_j\rangle_{s_j}$ ($k_j = 0, 1$), respectively, then the state of qubits $s_1', s_2', \cdots, s_n'$

will collapse into

$$|S'\rangle = [(1-t)(\alpha^*|0\rangle + \beta^*|1\rangle)_{s_1'} + t(\alpha|1\rangle - \beta|0\rangle)_{s_1'}] \otimes_{j=2}^n |k_j\rangle_{s_j'}. \tag{11}$$

After hearing Victor's measurement information, sender $A_1$ needs to performs a unitary operator $(-1)^{t+1}i\sigma_y$ on qubit $s_1$. Subsequently, sender $A_1$ and each $A_j$ ($j = 2, 3, \cdots, n$) jointly implement a unitary transformation $\mathcal{U}_j = (1 - k_j)U_1 + k_jU_2$ on the basis $\{|00\rangle_{s_1 s_j}, |01\rangle_{s_1 s_j}, |10\rangle_{s_1 s_j}, |11\rangle_{s_1 s_j}\}$ for particles $s_1$ and $s_j$, where $U_1$ and $U_2$ are given by

$$U_1 = \begin{pmatrix} 1 & 0 & 0 & 0 \\ 0 & 1 & 0 & 0 \\ 0 & 0 & 0 & 1 \\ 0 & 0 & 1 & 0 \end{pmatrix}, \quad U_2 = \begin{pmatrix} 0 & 1 & 0 & 0 \\ 1 & 0 & 0 & 0 \\ 0 & 0 & 1 & 0 \\ 0 & 0 & 0 & 1 \end{pmatrix}. \tag{12}$$

After executing the above operations by senders $A_1, A_2, \cdots, A_n$, the state $|S'\rangle$ shown in Eq (11) becomes

$$\begin{aligned} |S''\rangle &= (-1)^{t+1}i\sigma_y \otimes_{j=2}^n \mathcal{U}_j|S'\rangle \\ &= (1-t)(\alpha^* \otimes_{i=1}^n |1\rangle_{s_i'} - \beta^* \otimes_{i=1}^n |0\rangle_{s_i'}) \\ &\quad + t(\alpha \otimes_{i=1}^n |0\rangle_{s_i'} + \beta \otimes_{i=1}^n |1\rangle_{s_i'}) \\ &= (1-t)|S_\perp\rangle_s^n + t|S\rangle_s^n. \end{aligned} \tag{13}$$

Upon observing Victor's measurement outcome of qubit $s_1$ as $|\xi_1\rangle_{s_1}$ (i.e., $t = 1$) from Eq (13), it can be inferred that senders $A_1, A_2, \cdots, A_n$ are able to acquire a flawless replica of the collectively unknown state $|S\rangle_s^n$; otherwise (i.e., $t = 0$), they can obtain an orthogonal-complementing copy of the shared unknown state $|S\rangle_s^n$.

**Remark** (i) Due to the symmetry of qubits $s_1, s_2, \cdots, s_{n-1}$ and $s_n$ in state $|\phi^+\rangle^{\otimes n}$, Victor first measures anyone $s_j$ of qubits $s_1, s_2, \cdots, s_{n-1}$ and $s_n$ with the basis $\{|\xi_0\rangle, |\xi_1\rangle\}$, and then measures the other qubits with the $Z$ basis. The senders of the corresponding qubits perform the corresponding transformation in the above scheme, and the conclusion obtained is the same.

(ii) Regarding the issue of accessible information, it can be asserted that during the assisted cloning process, no subset can fully access quantum secrets. Now, let's take the teleportation process in the first stage of our scheme as an example to illustrate this conclusion. Assume that one sender $s_k$ attempts to reconstruct the secret at his or her location based on announced results by the other senders. For simplicity, let $m = 1$. Following the Bell-state measurement by all senders except $s_k$, the resulting state at $s_k$ is either $|\phi^-\rangle_{s_k}(\alpha|0\rangle + \beta|1\rangle)_r + |\phi^+\rangle_{s_k}(\alpha|0\rangle - \beta|1\rangle)_r$ or $|\psi^-\rangle_{s_k}(\alpha|0\rangle + \beta|1\rangle)_r + |\psi^+\rangle_{s_k}(\alpha|0\rangle - \beta|1\rangle)_r$. Upon tracing out the receiver's party, the reduced state at their end becomes $|\alpha|^2|00\rangle\langle00| + |\beta|^2|11\rangle\langle11|$ or $|\alpha|^2|01\rangle\langle01| + |\beta|^2|10\rangle\langle10|$. This holds true unless the entire channel is under their control, meaning that only amplitude information can be accessed by $s_j$. The same applies to any subparties of senders and receivers.

## 3 Assisted cloning of shared quantum secret via a non-maximally entangled GHZ-type state

The vulnerability of quantum entanglement and the inevitable impact of environmental noise can lead to the degeneration of a maximally entangled state into a non-maximally entangled

state. To deterministically obtain maximally entangled states, one can utilize quantum entanglement concentration and purification schemes [55–58], but, to achieve this, it is necessary to consume a large ensemble of non-maximally entangled states. Therefore, it is very meaningful to explore quantum communication problems by directly utilizing non-maximally entangled states as quantum channel.

Instead of sharing a $(n + m)$-qubit maximally entangled GHZ-type state as shown in Eq (2), the senders and receivers may initially share a non-maximally entangled state in the following form

$$|\mathcal{G}'\rangle_{n+m} = a \otimes_{l=1}^{n} |0\rangle_{s_l'} \otimes_{j=1}^{m} |0\rangle_{r_j} + b \otimes_{l=1}^{n} |1\rangle_{s_l'} \otimes_{j=1}^{m} |1\rangle_{r_j}, \tag{14}$$

where $a$ and $b$ are real numbers and satisfy $a^2 + b^2 = 1$. Without losing generality, assuming $|a| = \min\{|a|, |b|\}$. Using Bell state measurement bases, the initial composite system can be written as

$$
\begin{aligned}
|S\rangle_s^n |\mathcal{G}'\rangle_{n+m} &= \left(\alpha \otimes_{l=1}^{n} |0\rangle_{s_l} + \beta \otimes_{l=1}^{n} |1\rangle_{s_l}\right) \\
&\quad \otimes \left(a \otimes_{l=1}^{n} |0\rangle_{s_l'} \otimes_{j=1}^{m} |0\rangle_{r_j} + b \otimes_{l=1}^{n} |1\rangle_{s_l'} \otimes_{j=1}^{m} |1\rangle_{r_j}\right) \\
&= \frac{1}{\sqrt{2}} |\Phi_n^+\rangle_{s_1 \cdots s_n s_1' \cdots s_n'} \left(a\alpha \otimes_{j=1}^{m} |0\rangle_{r_j} + b\beta \otimes_{j=1}^{m} |1\rangle_{r_j}\right) \\
&\quad + \frac{1}{\sqrt{2}} |\Phi_n^-\rangle_{s_1 \cdots s_n s_1' \cdots s_n'} \left(a\alpha \otimes_{j=1}^{m} |0\rangle_{r_j} - b\beta \otimes_{j=1}^{m} |1\rangle_{r_j}\right) \\
&\quad + \frac{1}{\sqrt{2}} |\Psi_n^+\rangle_{s_1 \cdots s_n s_1' \cdots s_n'} \left(b\alpha \otimes_{j=1}^{m} |1\rangle_{r_j} + a\beta \otimes_{j=1}^{m} |0\rangle_{r_j}\right) \\
&\quad + \frac{1}{\sqrt{2}} |\Psi_n^-\rangle_{s_1 \cdots s_n s_1' \cdots s_n'} \left(b\alpha \otimes_{j=1}^{m} |1\rangle_{r_j} - a\beta \otimes_{j=1}^{m} |0\rangle_{r_j}\right),
\end{aligned} \tag{15}
$$

where $|\Phi^{\pm}\rangle_{s_1 \cdots s_n s_1' \cdots s_n'}$ and $|\Psi^{\pm}\rangle_{s_1 \cdots s_n s_1' \cdots s_n'}$ are shown in Eqs (5) and (6).

In the teleportation stage, each sender $A_j$ performs the standard Bell-state measurement on her or his two qubits $s_j$ and $s_j'$, and publish the measurement results to the receivers via classical communication. After performing $n$ times of Bell-state measurements, the state of qubits $r_1$, $r_2$, $\cdots$, $r_{m-1}$ and $r_m$ will collapse into one of the following four states

$$a\alpha \otimes_{j=1}^{m} |0\rangle_{r_j} \pm b\beta \otimes_{j=1}^{m} |1\rangle_{r_j}, \quad b\alpha \otimes_{j=1}^{m} |1\rangle_{r_j} \pm a\beta \otimes_{j=1}^{m} |0\rangle_{r_j}. \tag{16}$$

When the measurement outcome is $|\Phi^+\rangle_{s_1 \cdots s_n s_1' \cdots s_n'}$ or $|\Phi^-\rangle_{s_1 \cdots s_n s_1' \cdots s_n'}$, each receiver carries out the local Pauli operator $\sigma_z^X$ ($X = (\sum_{l=1}^{n} x_l) \bmod 2$), and the state of the receivers' particles becomes

$$|\dot{S}\rangle = \frac{1}{\sqrt{|a\alpha|^2 + |b\beta|^2}} \left(a\alpha \otimes_{j=1}^{m} |0\rangle_{r_j} + b\beta \otimes_{j=1}^{m} |1\rangle_{r_j}\right). \tag{17}$$

If the measurement result is $|\Psi^+\rangle_{s_1 \cdots s_n s_1' \cdots s_n'}$ or $|\Psi^-\rangle_{s_1 \cdots s_n s_1' \cdots s_n'}$, anyone of receivers executes the local Pauli operator $\sigma_x^Y \sigma_z^X$ ($Y = \sum_{l=1}^{n} y_l$), and the other receivers respectively carry out the

operator $\sigma_x^Y$, which will make the state owned by the receivers become

$$|\ddot{S}\rangle = \frac{1}{\sqrt{|b\alpha|^2 + |a\beta|^2}} (b\alpha \otimes_{j=1}^m |0\rangle_{r_j} + a\beta \otimes_{j=1}^m |1\rangle_{r_j}).$$ (18)

### 3.1 Assisted cloning based on projective measurement

The state corresponding to the measurement result $|\Phi_n^+\rangle$ or $|\dot{\Phi}_n^+\rangle$ is $|\dot{S}\rangle$, which is not yet the target state to be restored by receivers. In order to reconstruct the initial state with unity fidelity, an auxiliary qubit with the original state $|0\rangle_R$ is introduced. Due to the symmetry of the state $|\dot{S}\rangle$, any receiver $B_j$ can hold the auxiliary qubit. Without loss of generality, we can assume that the last receiver $B_m$ holds the auxiliary qubit and then performs a unitary operator

$$U_R = \begin{pmatrix} 1 & 0 & 0 & 0 \\ 0 & 1 & 0 & 0 \\ 0 & 0 & a/b & \sqrt{1 - a^2/b^2} \\ 0 & 0 & \sqrt{1 - a^2/b^2} & -a/b \end{pmatrix}$$ (19)

under the basis $\{|0\rangle_{r_m}|0\rangle_R, |0\rangle_{r_m}|1\rangle_R, |1\rangle_{r_m}|0\rangle_R, |1\rangle_{r_m}|1\rangle_R\}$. It will transform the product state $|\dot{S}\rangle \otimes |0\rangle_R$ to

$$\begin{aligned} U_R(|\dot{S}\rangle \otimes |0\rangle_R) &= \frac{a}{\sqrt{|a\alpha|^2 + |b\beta|^2}} (\alpha \otimes_{j=1}^m |0\rangle_{r_j} + \beta \otimes_{j=1}^m |1\rangle_{r_j})|0\rangle_R \\ &\quad + \frac{b\beta\sqrt{1 - a^2/b^2}}{\sqrt{|a\alpha|^2 + |b\beta|^2}} \otimes_{j=1}^m |1\rangle_{r_j})|1\rangle_R \\ &= \frac{a}{\sqrt{|a\alpha|^2 + |b\beta|^2}} |S\rangle_r^m |0\rangle_R + \frac{b\beta\sqrt{1 - a^2/b^2}}{\sqrt{|a\alpha|^2 + |b\beta|^2}} \otimes_{j=1}^m |1\rangle_{r_j})|1\rangle_R, \end{aligned}$$ (20)

where $|S\rangle_r^m = \alpha \otimes_{j=1}^m |0\rangle_{r_j} + \beta \otimes_{j=1}^m |1\rangle_{r_j}$ is shown in Section 2. The receiver $r_m$ then performs a $Z$-basis measurement on the auxiliary qubit $R$, which constitutes a projective measurement in the basis $\{|0\rangle, |1\rangle\}$. If the outcome is $|0\rangle_R$, the teleportation is successfully executed with fidelity 1, whereas if the outcome is $|1\rangle_R$, the teleportation fails without providing any information about the target state. The optimal probability of successful teleportation is $a^2$, where "optimal" refers to the introduction of an auxiliary qubit.

The measurement result $|\Psi_n^+\rangle$ or $|\dot{\Psi}_n^+\rangle$ necessitates the introduction of an auxiliary qubit with the original state $|0\rangle_{R'}$ at the position of the final receiver $B_m$, the corresponding unitary operator is

$$U_{R'} = \begin{pmatrix} a/b & \sqrt{1 - a^2/b^2} & 0 & 0 \\ \sqrt{1 - a^2/b^2} & a/b & 0 & 0 \\ 0 & 0 & 1 & 0 \\ 0 & 0 & 0 & 1 \end{pmatrix},$$ (21)

which will transform the product state $|\ddot{S}\rangle \otimes |0\rangle_{R'}$ to

$$
\begin{aligned}
U_{R'}(|\ddot{S}\rangle \otimes |0\rangle_{R'}) \quad &= \frac{a}{\sqrt{|b\alpha|^2 + |a\beta|^2}} (\alpha \otimes_{j=1}^m |0\rangle_{r_j} + \beta \otimes_{j=1}^m |1\rangle_{r_j})|0\rangle_{R'} \\[6pt]
&+ \frac{b\alpha\sqrt{1 - a^2/b^2}}{\sqrt{|b\alpha|^2 + |a\beta|^2}} \otimes_{j=1}^m |0\rangle_{r_j})|1\rangle_{R'} \\[6pt]
&= \frac{a}{\sqrt{|b\alpha|^2 + |a\beta|^2}} |S\rangle_r^m |0\rangle_{R'} + \frac{b\alpha\sqrt{1 - a^2/b^2}}{\sqrt{|b\alpha|^2 + |a\beta|^2}} \otimes_{j=1}^m |0\rangle_{r_j})|1\rangle_{R'}.
\end{aligned}
\tag{22}
$$

Next, the receiver $B_m$ performs a projective measurement with $Z$-basis on the auxiliary qubit $R'$. When the outcome is is $|1\rangle_{R'}$, the teleportation fails. If the measurement result is $|0\rangle_{R'}$, the target state can be reconstructed with a probability of $a^2$.

The optimal probability of successful teleportation is obtained by adding both contributions, resulting in $2a^2$. That is to say, the task of teleporting shared quantum secret in the first stage has been completed with a probability of $2a^2$.

The cloning of shared quantum secret state in the second stage is completely consistent with the Section 2, we will not repeat it here.

**Remark**: (i) When $a = b = 1/\sqrt{2}$, the success probability of our scheme is $2a^2 = 2 \times (1/\sqrt{2})^2 = 1$, and the quantum channel $|\mathcal{G}'\rangle_{n+m}$ shown in Eq (14) degenerates into the maximally entangled channel $|\mathcal{G}\rangle_{n+m}$ shown in Eq (2), which indicates that this scheme is a generalization of that in Section 2.

(ii) Note that the coefficients of quantum channel $|\mathcal{G}'\rangle_{n+m}$ are all real numbers. More generally, if the quantum channel is in the following form

$$
|\mathcal{G}''\rangle_{n+m} = a e^{i\theta_1} \otimes_{l=1}^n |0\rangle_{s'_l} \otimes_{j=1}^m |0\rangle_{r_j} + b e^{i\theta_2} \otimes_{l=1}^n |1\rangle_{s'_l} \otimes_{j=1}^m |1\rangle_{r_j},
\tag{23}
$$

where real numbers $a$, $b$, $\theta_1$ and $\theta_2$ satisfy $a^2 + b^2 = 1$ and $\theta_1, \theta_2 \in [0, 3\pi]$, then one of the receivers applies a unitary transformation $\tilde{U} = \mathrm{diag}(e^{-i\theta_1}, e^{-i\theta_2})$ under the computational basis ($Z$-basis) $\{|0\rangle, |1\rangle\}$ on his or her qubit, i.e.
$\tilde{U}|\mathcal{G}''\rangle_{n+m} = a \otimes_{l=1}^n |0\rangle_{s'_l} \otimes_{j=1}^m |0\rangle_{r_j} + b \otimes_{l=1}^n |1\rangle_{s'_l} \otimes_{j=1}^m |1\rangle_{r_j}$, which converts $\tilde{U}|\mathcal{G}''\rangle_{n+m}$ into $|\mathcal{G}'\rangle_{n+m}$. Therefore, applying the method in this subsection, the corresponding assisted cloning task can always be completed with a certain probability.

## 3.2 Assisted cloning based on positive operator-valued measurement

Let's only consider Eq (17), as the discussion on Eq (18) yields the same result. After introducing an auxiliary qubit $R$ with the original state $|0\rangle_R$ by the last receiver $B_m$, he or she performs a controlled-NOT gate $\mathcal{N}$ on qubits $r_m$ and $R$, where qubit $r_m$ works as the control qubit and qubit $R$ as the target qubit, and define $\mathcal{N}$ as $\mathcal{N}|lk\rangle = |l, (l + k)\mathrm{mod}2\rangle$. It will transform the product state $|\dot{S}\rangle \otimes |0\rangle_R$ to

$$
\begin{aligned}
\mathcal{N}(|\dot{S}\rangle \otimes |0\rangle_R) \quad &= \frac{1}{\sqrt{|a\alpha|^2 + |b\beta|^2}} (a\alpha \otimes_{j=1}^m |0\rangle_{r_j} |0\rangle_R + b\beta \otimes_{j=1}^m |1\rangle_{r_j} |1\rangle_R) \\[6pt]
&= \frac{1}{2\sqrt{|a\alpha|^2 + |b\beta|^2}} (|E\rangle_B \otimes |F\rangle_R + |G\rangle_B \otimes |H\rangle_R),
\end{aligned}
\tag{24}
$$

where

$$
\begin{aligned}
|E\rangle_B &= \alpha \otimes_{j=1}^m |0\rangle_{r_j} + \beta \otimes_{j=1}^m |1\rangle_{r_j}, \\
|F\rangle_R &= a|0\rangle_R + b|1\rangle_R, \\
|G\rangle_B &= \alpha \otimes_{j=1}^m |0\rangle_{r_j} - \beta \otimes_{j=1}^m |1\rangle_{r_j}, \\
|H\rangle_R &= a|0\rangle_R - b|1\rangle_R.
\end{aligned}
\tag{25}
$$

In order to determine $|E\rangle_B$ and $|G\rangle_B$, the receiver $B_m$ needs to perform an optimal positive operator-value measurement (POVM) on the auxiliary qubit $R$, which should take the following forms

$$
P_1 = \frac{1}{\lambda}|M_1\rangle\langle M_1|, \quad P_2 = \frac{1}{\lambda}|M_2\rangle\langle M_2|, \quad P_3 = I - \frac{1}{\lambda}\sum_{j=1}^2 |M_j\rangle\langle M_j|,
\tag{26}
$$

where $I$ is an identity operator, and

$$
\begin{aligned}
|M_1\rangle &= \frac{1}{\sqrt{\zeta}}\left(\frac{1}{a}|0\rangle_R + \frac{1}{b}|1\rangle_R\right), \\
|M_2\rangle &= \frac{1}{\sqrt{\zeta}}\left(\frac{1}{a}|0\rangle_R - \frac{1}{b}|1\rangle_R\right), \\
\zeta &= \frac{1}{a^2} + \frac{1}{b^2} = \frac{1}{a^2 b^2},
\end{aligned}
\tag{27}
$$

and the $\lambda$ related to parameters $a$ and $b$ should ensure that $P_3$ is a semi positive operator. To determine $\lambda$, we need to rewrite $P_1$, $P_2$ and $P_3$ in matrix form

$$
\begin{aligned}
P_1 &= \frac{1}{\lambda\zeta}\begin{pmatrix} a^{-2} & (ab)^{-1} \\ (ab)^{-1} & b^{-2} \end{pmatrix}, \\
P_2 &= \frac{1}{\lambda\zeta}\begin{pmatrix} a^{-2} & (ab)^{-1} \\ (ab)^{-1} & b^{-2} \end{pmatrix}, \\
P_3 &= \begin{pmatrix} 1 - 2(\lambda\zeta)^{-1}a^{-2} & 0 \\ 0 & 1 - 2(\lambda\zeta)^{-1}b^{-2} \end{pmatrix}.
\end{aligned}
\tag{28}
$$

To make $P_3$ a semi positive operator, the parameter $\lambda$ should satisfy the condition

$$
\lambda \geq 2\zeta^{-1}\max(a^{-2}, b^{-2}) = 2a^2 b^2 \max(a^{-2}, b^{-2}) = 2b^2.
$$

After executing POVM, receiver $B_m$ is able to obtain $P_j$ ($j = 1, 2$) with the following probability

$$
p(P_j) = \langle T|P_j|T\rangle = \frac{1}{\lambda\zeta} \quad (j = 1, 2),
\tag{29}
$$

where $|T\rangle = (a\alpha \otimes_{j=1}^m |0\rangle_{r_j}|0\rangle_R + b\beta \otimes_{j=1}^m |1\rangle_{r_j}|1\rangle_R)$. According to the value $\frac{1}{\lambda\mu}$ of POVM, receiver $B_m$ can infer the state $|F_j\rangle_R$ ($j = 1, 2$) of auxiliary qubit $R$. However, based on the value

$1 - \frac{2}{\lambda\zeta}$, receiver $B_m$ can obtain $U_5$, but cannot infer the states of qubit $R$. Once receiver $B_m$ determines the state $|F_j\rangle_R$ ($j = 1, 2$), it means he or she knows the state $|E_j\rangle_B$ ($j = 1, 2$), and then the receiver $B_m$ applies the corresponding unitary transformation $I$ or $\sigma_z$ to qubit $r_m$.

In this way, the state of qubits $r_1, r_2, \cdots, r_{m-1}$ and $r_m$ becomes $|S\rangle_r^m$ with the probability of $\frac{2}{\lambda\zeta}$, completing quantum teleportation.

In summary, the task in the first phase is completed with a probability of $4/\lambda\zeta$ and unit fidelity, because starting from Eq (18), the original state can also be reconstructed with a probability of $2/\lambda\zeta$ and unit fidelity.

Similar to Subsection 3.2, the auxiliary cloning of unknown shared quantum state in the second stage is completely consistent with the corresponding part in Section 2.

**Remark**: When $a = b = 1/\sqrt{2}$ and $\lambda = 1$, and the quantum channel $|\mathcal{G}'\rangle_{n+m}$ shown in Eq (14) changes into the maximally entangled channel $|\mathcal{G}\rangle_{n+m}$ shown in Eq (2), and the success probability of quantum teleportation in first stage is

$$\frac{4}{\lambda\zeta} = 4a^2b^2 = 4 \times \left(\frac{1}{\sqrt{2}}\right)^2 \times \left(\frac{1}{\sqrt{2}}\right)^2 = 1,$$

which means that the first stage here is standard quantum teleportation of shared secret. Combining with the second stage, our scheme here is a generalization of the scheme in Section 2.

### 3.3 Assisted cloning based on a single generalized Bell-state measurement

In the two preceding subsections, achieving 100% fidelity in teleporting the original shred state requires the introduction of an auxiliary qubit and subsequent execution of a two-qubit transformation. However, we demonstrate here that it is possible for receivers to restore the target state without an auxiliary qubit, albeit not with unit probability. This can be achieved by replacing $n$ Bell-state measurements with a single generalized Bell-state measurement. To begin, construct the generalized Bell-state basis as follows

$$|\phi_g^+\rangle = b|00\rangle + a|11\rangle, \quad |\phi_g^-\rangle = a|00\rangle - b|11\rangle,$$

$$|\psi_g^+\rangle = b|01\rangle + a|10\rangle, \quad |\psi_g^-\rangle = a|01\rangle - b|10\rangle, \tag{30}$$

and use this generalized Bell-state measurement basis to rewrite $|S\rangle_s^n |\mathcal{G}'\rangle_{n+m}$ as follows

$$
\begin{aligned}
|S\rangle_s^n |\mathcal{G}'\rangle_{n+m} =\ & |\phi_g^+\rangle_{s_1,s_1'} |\Phi_{n-1}^+\rangle_{s_2,s_2',\cdots,s_n,s_n'} ab\left(\alpha \otimes_{j=1}^m |0\rangle_{r_j} + \beta \otimes_{j=1}^m |1\rangle_{r_j}\right) \\
& + |\phi_g^+\rangle_{s_1,s_1'} |\Phi_{n-1}^-\rangle_{s_2,s_2',\cdots,s_n,s_n'} ab\left(\alpha \otimes_{j=1}^m |0\rangle_{r_j} - \beta \otimes_{j=1}^m |1\rangle_{r_j}\right) \\
& + |\phi_g^-\rangle_{s_1,s_1'} |\Phi_{n-1}^+\rangle_{s_2,s_2',\cdots,s_n,s_n'} \left(a^2\alpha \otimes_{j=1}^m |0\rangle_{r_j} - b^2\beta \otimes_{j=1}^m |1\rangle_{r_j}\right) \\
& + |\phi_g^-\rangle_{s_1,s_1'} |\Phi_{n-1}^-\rangle_{s_2,s_2',\cdots,s_n,s_n'} \left(a^2\alpha \otimes_{j=1}^m |0\rangle_{r_j} + b^2\beta \otimes_{j=1}^m |1\rangle_{r_j}\right) \\
& + |\psi_g^+\rangle_{s_1,s_1'} |\Psi_{n-1}^+\rangle_{s_2,s_2',\cdots,s_n,s_n'} \left(b^2\alpha \otimes_{j=1}^m |1\rangle_{r_j} + a^2\beta \otimes_{j=1}^m |0\rangle_{r_j}\right) \\
& + |\psi_g^+\rangle_{s_1,s_1'} |\Psi_{n-1}^-\rangle_{s_2,s_2',\cdots,s_n,s_n'} \left(b^2\alpha \otimes_{j=1}^m |1\rangle_{r_j} - a^2\beta \otimes_{j=1}^m |0\rangle_{r_j}\right) \\
& + |\psi_g^-\rangle_{s_1,s_1'} |\Psi_{n-1}^+\rangle_{s_2,s_2',\cdots,s_n,s_n'} ab\left(\alpha \otimes_{j=1}^m |1\rangle_{r_j} + \beta \otimes_{j=1}^m |0\rangle_{r_j}\right) \\
& + |\psi_g^-\rangle_{s_1,s_1'} |\Psi_{n-1}^-\rangle_{s_2,s_2',\cdots,s_n,s_n'} ab\left(\alpha \otimes_{j=1}^m |1\rangle_{r_j} - \beta \otimes_{j=1}^m |0\rangle_{r_j}\right),
\end{aligned}
\tag{31}
$$

where $|\Phi_{n-1}^+\rangle$, $|\Phi_{n-1}^-\rangle$, $|\Psi_{n-1}^+\rangle$ and $|\Psi_{n-1}^-\rangle$ are defined by Eq (6). Here, the sender $A_1$ executes a

generalized Bell-state measurement on qubit pair $(s_1, s'_1)$. In fact, any sender in the group is capable of conducting the generalized Bell-state measurement due to the symmetry of the original state $|S\rangle^n_s$ and the network channel $|\mathcal{G}'\rangle_{n+m}$, resulting in identical outcomes. Without loss of generality, let's assume that the first sender is responsible for this operation, while the remaining senders continue with standard Bell state measurements. It is evident that the outcome remains unaffected by the order in which joint measurements are conducted.

If the measurement outcome is $|\phi_g^+\rangle|\Phi_{n-1}^\pm\rangle$, the state at the receivers' hands will be $(\alpha \otimes_{j=1}^m |0\rangle_{r_j} \pm \beta \otimes_{j=1}^m |1\rangle_{r_j})$, and when the result is $|\psi_g^-\rangle|\Psi_{n-1}^\pm\rangle$, the state will be $(\alpha \otimes_{j=1}^m |1\rangle_{r_j} \pm \beta \otimes_{j=1}^m |0\rangle_{r_j})$. In both of these scenarios, the receivers have the capability to restore the target state by implementing appropriate local Pauli operators. Consequently, the probability of successful restoration is denoted as $p_1 = 2a^2$, where $p_1$ represents the likelihood of success without the introduction of an auxiliary qubit.

If the measurement results are $|\phi_g^-\rangle|\Phi_{n-1}^\pm\rangle$ or $|\phi_g^+\rangle|\Psi_{n-1}^\pm\rangle$, the unnormalized state at the receivers' hands will be $(a^2\alpha \otimes_{j=1}^m |0\rangle_{r_j} \mp b^2\beta \otimes_{j=1}^m |1\rangle_{r_j})$ or $(b^2\alpha \otimes_{j=1}^m |1\rangle_{r_j} \pm a^2\beta \otimes_{j=1}^m |0\rangle_{r_j})$, respectively. The receivers can obtain the target state similar to the scheme in Subsection 3.1 by introducing an auxiliary state and finding the corresponding general evolution separately through replacing $(a, b)$ with $(a^2, b^2)$ in Eq (16). As a result, the optimal successful probability is $p_2 = 2a^2$, where $p_2$ represents the successful probability when introducing an auxiliary qubit.

Note that the probability of randomly obtaining any one of $|\phi_g^+\rangle|\Phi_{n-1}^\pm\rangle$, $|\psi_g^-\rangle|\Psi_{n-1}^\pm\rangle$, $|\phi_g^-\rangle|\Phi_{n-1}^\pm\rangle$ or $|\phi_g^+\rangle|\Psi_{n-1}^\pm\rangle$ and $|\psi_g^-\rangle|\Psi_{n-1}^\pm\rangle$ is 1/4, therefore the total success probability of quantum teleportation in the first stage is

$$\frac{1}{4} \times 2 \times (2a^2) + \frac{1}{4} \times 2 \times (2a^2) = 2a^2.$$

Now, let's consider the second stage of the scheme. By replacing state $(|\phi^+\rangle)^{\otimes n}$ in Section 2 with state $|\phi_g^+\rangle(|\phi^+\rangle)^{\otimes(n-1)}$, the state corresponding to Eq (10) in Section 2 is

$$|\phi_g^+\rangle(|\phi^+\rangle)^{\otimes(n-1)} = \frac{1}{\sqrt{2^n}}[|\xi_0\rangle_{s_1}(b\alpha^*|0\rangle + a\beta^*|1\rangle)_{s'_1}$$

$$+ |\xi_1\rangle_{s_1}(a\alpha|1\rangle - b\beta|0\rangle)_{s'_1}] \otimes_{i=2}^n \sum_{k_i=0}^1 |k_i\rangle_{s_i}|k_i\rangle_{s'_i}. \tag{32}$$

After the state preparer Victor and the senders perform the same operations as the second stage of the scheme in Section 2, the state corresponding to Eq (13) in Section 2 is

$$|\bar{S}''\rangle = (1-t)(b\alpha^* \otimes_{i=1}^n |1\rangle_{s'_i} - a\beta^* \otimes_{i=1}^n |0\rangle_{s'_i})$$

$$+ t(a\alpha \otimes_{i=1}^n |0\rangle_{s'_i} + b\beta \otimes_{i=1}^n |1\rangle_{s'_i}). \tag{33}$$

That is to say, the states corresponding to Victor's measurement results $|\xi_0\rangle_{s_1}$ and $|\xi_1\rangle_{s_1}$ are the states $b\alpha^* \otimes_{i=1}^n |1\rangle_{s'_i} - a\beta^* \otimes_{i=1}^n |0\rangle_{s'_i}$ and $a\alpha \otimes_{i=1}^n |0\rangle_{s'_i} + b\beta \otimes_{i=1}^n |1\rangle_{s'_i}$, respectively.

Finally, when sender $A_1$ uses the methods described in Subsections 3.1 or 3.2, all senders can obtain a copy or an orthogonal-complementing copy of the unknown shared state with probability $a^2$.

For other measurement results in the process of quantum teleportation (see Eq (31)), applying the same analysis method as above, senders will obtain a copy or an orthogonal-complementing copy of the unknown shared state with a certain probability.

**Remark**: (i) The construction method of generalized Bell-state basis is not unique. For example, the vectors $|\phi_g^+\rangle = a|00\rangle + b|11\rangle$, $|\phi_g^-\rangle = b|00\rangle - a|11\rangle$, $|\psi_g^+\rangle = a|01\rangle + b|10\rangle$ and $|\psi_g^-\rangle = b|01\rangle - a|10\rangle$ also form a set of mutually orthogonal generalized Bell-state bases.

(ii) Similar to the discussion in Subsections 3.1 or 3.2, this scheme is still a generalization of the scheme in Section 2.

## 4 Discussion and conclusion

As the first stage of our scheme, quantum teleportation is different from the design concept in references [59, 68] in which a trusted node plays an important role in connecting participants and transmitting information. Establishing a long-distance quantum communication through distributed nodes is advantageous, as it eliminates the need for any single node to relay complete quantum information. This principle also extends to the storage and retrieval of quantum secrets in spatially separated quantum memory. Verification strategies for multipartite entanglement [58–61] are valuable for preparing an entangled network in the presence of untrustworthy parties. The access information in schemes with non-maximally entangled channels can be displayed using the same method as described in Subsection 3.1 of this article. In this way, we still have the conclusion that no subparties can fully access quantum secrets during the process of teleportation.

Note that except for the $(n-1)$ times of standard Bell-state measurements used in Subsection 3.3, all other schemes used $n$ times of standard Bell-state measurements, which are unnecessary. Actually, only one Bell-state measurement or generalized Bell-state measurement is sufficient for implemening each scheme. Rewrite Eq (6) as (wothout normalized)

$$
\begin{aligned}
|\Phi_n^+\rangle &= |\phi^+\rangle \sum_{0 \le k = 0 \bmod 2 \le n} N[|-\rangle^{\otimes k}|+\rangle^{\otimes (2n-2-k)}] \\
&\quad + |\phi^-\rangle \sum_{0 \le k = 1 \bmod 2 \le n} N[|-\rangle^{\otimes k}|+\rangle^{\otimes (2n-2-k)}], \\
|\Phi_n^-\rangle &= |\phi^+\rangle \sum_{0 \le k = 1 \bmod 2 \le n} N[|-\rangle^{\otimes k}|+\rangle^{\otimes (2n-2-k)}] \\
&\quad + |\phi^-\rangle \sum_{0 \le k = 0 \bmod 2 \le n} N[|-\rangle^{\otimes k}|+\rangle^{\otimes (2n-2-k)}], \\
|\Psi_n^+\rangle &= |\psi^+\rangle \sum_{0 \le k = 0 \bmod 2 \le n} N[|-\rangle^{\otimes k}|+\rangle^{\otimes (2n-2-k)}] \\
&\quad + |\psi^-\rangle \sum_{0 \le k = 1 \bmod 2 \le n} N[|-\rangle^{\otimes k}|+\rangle^{\otimes (2n-2-k)}], \\
|\Psi_n^-\rangle &= |\psi^+\rangle \sum_{0 \le k = 1 \bmod 2 \le n} N[|-\rangle^{\otimes k}|+\rangle^{\otimes (2n-2-k)}] \\
&\quad + |\psi^-\rangle \sum_{0 \le k = 0 \bmod 2 \le n} N[|-\rangle^{\otimes k}|+\rangle^{\otimes (2n-2-k)}],
\end{aligned}
\tag{34}
$$

implying that executing one Bell-state measurement and $2(n-1)$ times of $X$-basis measurements is equivalent to $n$ times of Bell-state measurements (or one generalized Bell-state measurement and $(n-1)$ times of Bell-state measurements), where the $X$-basis measurement is a single-qubit projective measurement on the basis $\{|\pm\rangle = (|0\rangle \pm |1\rangle)/\sqrt{2}\}$. Due to the greater experimental feasibility of executing two single-qubit measurements compared to a joint two-

qubit measurement, it appears that the latter option, involving only one Bell-state (generalized Bell-state) measurement, should be chosen. The success probability of identifying the Bell states $|\phi^-\rangle$ and $|\psi^-\rangle$ is limited to 1/2, as only these two states can be unambiguously distinguished from each other [62, 63]. When performing $n$ Bell-state measurements according to Eq (6), failure occurs only when the measurement result is $|\phi^+\rangle^{\otimes n}$ or $|\psi^+\rangle^{\otimes n}$. As a result, the probability of successful discrimination increases to $1 - 2^{-n}$, indicating that increasing $n$ enhances the likelihood of successfully distinguishing between the logical Bell states.

Of course, our work is not limited to the GHZ-type state encoding. By correcting photon loss, operational errors, and dishonest participants through error coding, it can be further extended to fault tolerance. For instance, a parity state encoding [64] can be used to some extent [65, 66] to correct the effects of photon loss, errors, and dishonesty. In principle, even in the case of loss and error, it can transmit quantum information with any high probability of success [67]. The use of other types of entangled states such as cluster states for encoding is worth further consideration. The success probability of Bell state measurement can be enhanced by utilizing cluster state encoding [68]. The integration of such encoding techniques and secret sharing protocols [11] based on cluster states is of great significance. In addition, there is only one state preparer in our schemes. We can introduce two or more state preparers like in article [69], which can improve the security of our scheme. To achieve this, we need to introduce an appropriate amount of auxiliary particles and use joint RSP technology.

It is worth noting that our scheme has high security. To make it clearer, we give a security check here. Before initiating bidirectional controlled assisted cloning, they should first conduct a security check. Alice prepares one check sequence composed of qudits with the random state $\{|0\rangle, |1\rangle, |+\rangle, |-\rangle\}$ and sends it to the remote preparer Victor. When the eavesdropper Eve intercepted this qudit sequence, he will randomly select a set of polarization-based measurements $\{|0\rangle, |1\rangle\}$ or $\{|+\rangle, |-\rangle\}$ to measure the check qudits, and prepares a new qudit sequence to Victor. However, due to Eve's behavior interfering with quantum states, if he chooses the wrong measurement basis, it will lead to a high error rate. This high error rate will be detected during the process of security check between Alice and Victor. If there is one eavesdropper, it is possible for Alice and Victor to suspend the communication. The same security detection method can be used between Bob and Victor, as well as among Alice, Bob and Charlie. Therefore, the security of our scheme can be guaranteed. On the other hand, due to the introduction of a controller in our scheme, the security of the scheme has been further enhanced.

In summary, we have proposed a new protocol that one can produce perfect copies and orthogonal complement copies of an arbitrary unkonwn shared quantum state via quantum and classical channel, with assistance of a state preparer. This assisted cloing ptotocol needs two stage. The first stage requires teleportation by using a maximally entangled GHZ-type state as quantum channel to teleport an arbitrary unknown shared quantum state between multiple parties in a quantum network. In the second stage, the state preparer executes a special single-qubit projective measurement and a series of single-qubit computational basis measurements on the qubits which seeded by senders. After having received the preparer's measurement outcomes through classical channel, senders can obtain the input original state and its orthogonal complement state by a series of appropriate unitary operations. In order to meet the needs of the real environment, we have extended the above protocols to the case of non-maximally entangled GHZ-type quantum network channel, and obtained three generalized protocols with unit fidelity a certain probability. In the first generalized protocol, one of the receivers needs to introduce an auxiliary qubit, perform a twoqubit unitary transformation, and make a single-qubit computational basis measurement on the auxiliary qubit. The second generalized protocol requires one of the receivers to perform a controlled-NOT gate transformation and POVM after introducing an auxiliary qubit. The first stage of the third generalized

protocol requires one of the senders to perform a generalized Bell state measurement. After other senders perform standard Bell measurements, the receivers either directly recover the input unknown state through appropriate Palui gates, or probabilistically reconstruct the target state using the method of the first or second generalized protocol. In the second stage of this protocol, the senders need to use the receivers' method in the first or second generalized protocols to complete the cloning task.

## Author Contributions

**Writing – original draft:** Dengxin Zhai, Jiayin Peng.

**Writing – review & editing:** Dengxin Zhai, Jiayin Peng, Nueraminaimu Maihemuti, Jiangang Tang.

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
