## [Decision Letter · Decision Letter 0]

4 Jun 2024

Assisted cloning of an unknown shared quantum state

PONE-D-24-03893

Dear Dr. Peng,

We’re pleased to inform you that your manuscript has been judged scientifically suitable for publication and will be formally accepted for publication once it meets all outstanding technical requirements.

Kind regards,

Salvatore Lorenzo

Academic Editor

PLOS ONE

Journal Requirements:

Quantum Teleportation of Shared Quantum Secret - https://physics.snu.ac.kr/hjeong/pdf/2020/PhysRevLett.124.060501.pdf

Probabilistic quantum teleportation of shared quantum secret - 10.1088/1674-1056/ac67c0

In your revision ensure you cite all your sources (including your own works), and quote or rephrase any duplicated text outside the methods section. Further consideration is dependent on these concerns being addressed.

"This work is supported by the Kashi University Flexible Introduction Research Initiation Fund (No. 022024077)"

Please respond by return e-mail so that we can amend your financial disclosure and competing interests on your behalf.

4. Please update your submission to use the PLOS LaTeX template. The template and more information on our requirements for LaTeX submissions can be found at http://journals.plos.org/plosone/s/latex.

Additional Editor Comments (optional):

Reviewers' comments:

Reviewer's Responses to Questions

**Comments to the Author**

1. Is the manuscript technically sound, and do the data support the conclusions?

Reviewer #1: Yes

2. Has the statistical analysis been performed appropriately and rigorously? 

Reviewer #1: Yes

3. Have the authors made all data underlying the findings in their manuscript fully available?

Reviewer #1: Yes

4. Is the manuscript presented in an intelligible fashion and written in standard English?

Reviewer #1: Yes

5. Review Comments to the Author

Reviewer #1: This paper proposes some innovative quantum information processing schemes, focusing on the challenge of assisted cloning of unknown shared state. The authors propose a new scheme to implement quantum cloning of an arbitrary unknown shared state with assistance of a state preparer. The first stage of this scheme requires teleportation, which allows the quantum state shared by an arbitrary number of senders to be teleported to another arbitrary number of receivers via a maximally entangled GHZ-type state as quantum channel. In the second stage of this scheme, the state preparer executes some projective measurements and announces measurement results, the perfect copy or orthogonal-complementing copy of an unknown shared state can be produced. In addition, the proposed scheme is extended to the case of non-maximally entangled GHZ-type network channel from three aspects: projective measurement, POVM and generalized Bell measurement. The paper presents a clear hierarchy, and its innovative methods in theoretical framework and scheme design are particularly commendable. This work significantly expands and deepens existing theoretical understanding, and also has considerable foresight. Overall, this article demonstrates the enormous potential and academic value of its field, with the following five important contributions or significance:

(1) Each of the proposed schemes includes two parts: teleportation and assisted cloning. As is well known, a teleportation between multiple senders and receivers has been rare so far. As far as I know, there are few previous schemes that allow people to transfer shared quantum information among multiple parties directly to others without concentrating the information in the location of single or subparties. This article is one of the exemplary schemes to overcome the above drawbacks. On the other hand, existing auxiliary cloning schemes only clone quantum states at the position of the unique sender. This article can jointly clone a quantum state in the respective locations of multiple senders, which interprets the concept of quantum cloning from a completely different perspective, overturning people's understanding of existing assisted cloning.

(2) The proposed schemes can relay quantum information over a network in an efficient and distributed manner without requiring fully trusted central or intermediate nodes. It can be further extended to include error corrections against photon losses, bit or phase-flip errors, and dishonest parties. This work thus opens a route to the realizations of secure distributed quantum communications and computations in quantum networks.

(3) Discussing the cloning via non-maximally entangled channel means expanding the cloning scheme in ideal environments to noisy environments. This multi-party assisted cloning scheme lays a solid foundation for real-world applications. In addition, this article expands the cloning scheme in ideal environments from three different measurement methods, showcasing the intersection and integration of various ideas, methods, theories and quantum technologies. It greatly expands people's perspective on designing flexible and versatile quantum communication protocols by comprehensively utilizing various knowledge and technologies.

(4) The proposed scheme provides a mathematical analysis method about how no sub part can fully access quantum secrets during the process of assisted cloning. This method is of great significance for inspiring people to evaluate the security of quantum communication protocols.

(5) The analysis of the possible encoding channels, number of preparers, the relationship between Bell measurement and X-based measurement, safety of the cloning scheme enhances the theoretical depth of the proposed schemes, which not only makes it safer and more flexible, but also greatly expands the application space of the scheme in various scenarios.

Based on the above reasons, I am willing and strongly recommend accepting its publication.

6. PLOS authors have the option to publish the peer review history of their article (what does this mean?). If published, this will include your full peer review and any attached files.

Reviewer #1: No

---

## [Editor Report · Acceptance letter]

16 Aug 2024

PONE-D-24-03893 

PLOS ONE

Dear Dr. Peng, 

I'm pleased to inform you that your manuscript has been deemed suitable for publication in PLOS ONE. Congratulations! Your manuscript is now being handed over to our production team.

Kind regards, 

on behalf of

Dr. Salvatore Lorenzo 

Academic Editor

PLOS ONE